# Phytochemical Analysis, Biological Activities, and Docking of Phenolics from Shoot Cultures of *Hypericum perforatum* L. Transformed by *Agrobacterium rhizogenes*

**DOI:** 10.3390/molecules29163893

**Published:** 2024-08-17

**Authors:** Oliver Tusevski, Marija Todorovska, Jasmina Petreska Stanoeva, Sonja Gadzovska Simic

**Affiliations:** 1Institute of Biology, Faculty of Natural Sciences and Mathematics, Ss. Cyril and Methodius University in Skopje, 1000 Skopje, North Macedonia; marija.todorovska@pmf.ukim.mk; 2Institute of Chemistry, Faculty of Natural Sciences and Mathematics, Ss. Cyril and Methodius University in Skopje, 1000 Skopje, North Macedonia; jasmina.petreska@pmf.ukim.mk

**Keywords:** biological activities, *Hypericum perforatum*, molecular docking, phenolic compounds, transformed shoots

## Abstract

*Hypericum perforatum* transformed shoot lines (TSL) regenerated from corresponding hairy roots and non-transformed shoots (NTS) were comparatively evaluated for their phenolic compound contents and in vitro inhibitory capacity against target enzymes (monoamine oxidase-A, cholinesterases, tyrosinase, α-amylase, α-glucosidase, lipase, and cholesterol esterase). Molecular docking was conducted to assess the contribution of dominant phenolic compounds to the enzyme-inhibitory properties of TSL samples. The TSL extracts represent a rich source of chlorogenic acid, epicatechin and procyanidins, quercetin aglycone and glycosides, anthocyanins, naphthodianthrones, acyl-phloroglucinols, and xanthones. Concerning in vitro bioactivity assays, TSL displayed significantly higher acetylcholinesterase, tyrosinase, α-amylase, pancreatic lipase, and cholesterol esterase inhibitory properties compared to NTS, implying their neuroprotective, antidiabetic, and antiobesity potential. The docking data revealed that pseudohypericin, hyperforin, cadensin G, epicatechin, and chlorogenic acid are superior inhibitors of selected enzymes, exhibiting the lowest binding energy of ligand–receptor complexes. Present data indicate that *H. perforatum* transformed shoots might be recognized as an excellent biotechnological system for producing phenolic compounds with multiple health benefits.

## 1. Introduction

*Hypericum perforatum* L. (St. John’s wort) is the most investigated and exploited medicinal plant worldwide. The medicinal properties of *H. perforatum* include antidepressant, neuroprotective, antioxidant, antimicrobial, anti-inflammatory, anticancer, antidiabetic, and antihyperlipidemic activities [1,2]. These health benefit effects have been related to different groups of phenolic compounds, such as naphthodianthrones (hypericin and pseudohypericin), phloroglucinols (hyperforin and adhyperforin), flavonols (quercetin, rutin, hyperoside, quercitrin, and kaempferol), catechins (catechin, epicatechin, and procyanidin B2), phenolic acids (chlorogenic acid), and xanthones [2,3]. In this context, hypericins and hyperforins have been proposed as efficient antidepressant compounds through various mechanisms of action involving monoamine oxidase inhibition, inhibition of synaptosomal reuptake of neurotransmitters, as well the effects on monoamine transporters and serotonin receptors [4]. Flavonoid glycosides and aglycones from *H. perforatum* have attracted much interest as therapeutic compounds for many chronic diseases due to their well-documented antioxidant and free radical scavenging activities [5]. Recently, increasing attention has been paid to xanthones from *H. perforatum* as antihyperglycemic and insulintropic compounds through the inhibition of α-amylase and α-glucosidase, as well as the regulation of AMP-activated protein kinase expression and protein kinase C concentration [1,6]. In spite of this knowledge, studies elucidating the mechanism of action of phenolic compounds from *H. perforatum* responsible for some of the ascribed pharmacological properties are still lacking in the literature.

The *H. perforatum* raw material derived from wild-growing and field-cultivated *H. perforatum* plants has been significantly exploited by the pharmaceutical and food industry for the preparation of commercial remedies and dietary supplements [7]. However, open field conditions could induce significant variations in the yield and phytochemical composition of *H. perforatum* harvested biomass [2]. In this context, various environmental factors have been proposed as major components that influence the secondary metabolism of *Hypericum* sp., leading to inconsistent contents of bioactive metabolites [7]. Thus, the implementation of advanced biotechnological methods for plant production in controlled conditions could be of great interest to industries to obtain standardized *H. perforatum* formulations [3].

Plant cell and tissue culture has been considered as an efficient technology for the large-scale production of *H. perforatum* biomass with uniform production of bioactive compounds. In vitro shoots and roots of *H. perforatum* as differentiated cultures have been selected as perspective systems for the production of phenolics, flavonoids, hypericins, hyperforins, and xanthones [3]. Nevertheless, it has been shown that the type of explants, culture conditions, and phytohormone supplementation might cause significant variations in the metabolic profile of *H. perforatum* shoot and root cultures [8,9,10]. Concerning callus and cell cultures of this species, their inferior capacity for secondary metabolite biosynthesis has been related to their undifferentiated nature, as well as the lack of multicellular glandular structures as the main accumulation sites for hypericins and hyperforins [10,11]. Considering these findings, *H. perforatum* in vitro cultures have been subjected to various biotechnological tools for enhanced production of target compounds, such as elicitation and bioreactor cultivation [12,13]. Even though those strategies have shown promising outcomes for increased production of secondary metabolites, inconsistency in the increment of biomass and some desired metabolites are the main constraints that limit the commercial application of *H. perforatum* cultures [14].

Plant genetic transformation has been confirmed as a powerful methodology for biosynthetic capability improvement in *H. perforatum* and other species from the genus *Hypericum* [14]. Among the genetic modification methods, only *Agrobacterium rhizogenes*- and biolistic-mediated transformations have been established in *Hypericum* spp. [15,16,17,18,19]. On the other hand, *H. perforatum* has been shown to be a recalcitrant species to *Agrobacterium tumefaciens*-mediated genetic transformation due to its antibacterial properties [20]. However, most of the studies have been focused on the establishment of hairy roots (HR) by successful transfer of T-DNA *rol* and *aux* genes from the Ri-plasmid of *A. rhizogenes* into the *Hypericum* genome [15,17,18]. It has been reported that the integration of bacterial T-DNA genes into the plant genome may induce modification in the phytohormone signal perception and biochemical processes in transformed cells that result in the up-regulation of secondary metabolism [21]. Previous reports have revealed that *Hypericum* HR cultures represent a sustainable source of biomass enriched in hydroxybenzoic and hydroxycinnamic acids, monomeric and oligomeric catechins, flavonoids, and xanthones [18,19,22,23].

In our recent studies, we have screened fifteen dark-grown *H. perforatum* HR clones induced by *A. rhizogenes* strain A4 for total production of phenolic compounds and antioxidant status and have selected three clones for their detailed phytochemical profile and enzyme-inhibitory properties [22,24]. On the basis of in vitro and in silico analyses, we have revealed that selected HR clones enriched in xanthones possess a strong antidiabetic potential [22], which was further confirmed by in vivo investigation, implying that xanthones from *H. perforatum* HR regulate carbohydrate metabolism and blood glucose levels in diabetic rats [25]. Even though *H. perforatum* dark-grown HR clones were confirmed as a powerful source of xanthones as bioactive compounds, those cultures have not shown a capability for the production of naphthodianthrones and acyl-phloroglucinols, which are the most desirable metabolites for the pharmaceutical industry due to their multiple health benefits [7].

We continued our research towards the exposition of those fifteen *H. perforatum* HR lines to photoperiod conditions in order to select green HR clones with a newly acquired capability for the production of aerial part-specific compounds [26,27]. Even though selected photoperiod-exposed HR clones exhibited a potential for the production of pseudohypericin and protopseudohypericin, the minor amounts observed in the root samples was the main reason to exclude these naphthodianthrones for further in silico analysis and their contribution to the in vitro bioactivity assays [27]. One of the main achievements in those studies was the spontaneous regeneration of photoperiod-exposed HR clones into corresponding transformed shoot lines (TSL) that were previously evaluated for their growth characteristics, phenylpropanoid and naphthodianthrone production, as well as non-enzymatic and enzymatic antioxidant activities [28]. Noteworthily, we have observed a wide variation of hypericin, pseudohypericin, and protopseudohypericin contents among fifteen tested TSL (TSL A-TSL O), which were shown to be superior producers of naphthodianthrones compared to non-transformed shoots (NTS). The screening of these lines also revealed that TSL F with the slowest growth rate showed the strongest accumulation of total phenolic compounds and superior antioxidant activity, while TSL B with the best growth showed moderate production of phenolic compounds and antioxidant status [28]. All the previous experimental data forced us to extend the metabolic profiling of TSL B and TSL F with the opposite growth traits, one randomly selected line TSL H, as well as NTS of *H. perforatum* towards identifying phenolic compounds that were not previously observed in HR cultures and evaluating their contribution to the in vitro biological activities.

Taking into account our continuing endeavor to search for phytopharmaceuticals for the management of chronic diseases, this study was designed to evaluate:(1)phenolic profile (phenolic acids, flavan-3-ols, flavonols, anthocyanins, naphthodianthrones, acyl-phloroglucinols, and xanthones) in *H. perforatum* TSL and NTS extracts using HPLC/DAD/ESI-MS^n^ methodology;(2)enzyme-inhibitory activity against monoamine oxidase-A (MAO-A), acetylcholinesterase (AChE), butyrylcholinesterase (BChE), tyrosinase (TYR), α-amylase (α-AM), α-glucosidase (α-GL), pancreatic lipase (PL) and cholesterol esterase (CHE) by in vitro assays; and(3)potential interactions between representative phenolic compounds and target enzymes using molecular docking studies.

The outgoing results could provide a new insight into *H. perforatum* transformed shoots as an efficient biotechnological system for the production of bioactive compounds with pharmacological applications.

## 2. Results

### 2.1. HPLC/DAD/ESI-MS^n^ Analysis of Phenolic Compounds in H. perforatum Transformed Shoots

Phenolic acids. Chlorogenic acid (F2), 3-*p*-coumaroylquinic acid (F3), and 3-feruloylquinic acid (F5) were confirmed in TSL and NTS extracts (Table 1). Among these compounds, chlorogenic acid was found to be a dominant phenolic acid in tested TSL extracts. The TSL F showed a significantly higher F2 amount (1.4-fold) in comparison to NTS. In contrast, TSL B and TSL H exhibited a markedly lower F2 content (11.4- and 2.2-fold, respectively) than control shoots. All the TSL showed a significantly decreased F3 content (from 2.0- to 9.4-fold) compared to the NTS. Concerning F5, TSL H showed a 1.4-fold higher content, while TSL F displayed a 3-fold lower amount than control shoots.

Flavan-3-ols. The chromatographic analysis confirmed four flavan-3-ols (F1, F4, F6, and F7) in shoot extracts (Table 1). Among the identified flavan-3-ols, only F7 (epicatechin) was detected as the preeminent compound in all shoot samples. However, significant variation in F7 content between NTS and TSL was not observed, except in the TSL H clone exhibiting a slightly declined production of epicatechin (1.2-fold) compared to the control shoots. The compound F4 (procyanidin B2) was de novo synthesized in all the tested TSL and its amount was 1.8-fold higher in TSL F compared to TSL B and TSL H. Other identified flavan-3-ols, such as F1 (epicatechin-epigallocatechin dimer) and F6 (procyanidin trimer) were de novo produced in significant amounts only in TSL F. Among the tested clones, the TSL F was shown to be the richest source of flavan-3-ols.

Flavonols. The HPLC analysis showed the presence of five flavonol glycosides (F9, F11–F14) and one flavonoid aglycone (F15) in the analyzed shoot samples (Table 1). Two flavonol glycosides, such as hyperoside (F12) and quercitrin (F14), as well as flavonoid aglycone quercetin (F15), were detected in all TSL and NTS. The F12 and F14 contents in TSL were comparable or even lower compared to those observed in control shoots. Noteworthily, all the tested TSL exhibited significantly higher F15 amounts (from 1.3- to 1.7-fold) in comparison to NTS cultures. The compound F9 identified as quercetin 6-*C*-glucoside was confirmed only in NTS and TSL F. The component F11 (kaempferol 6-*C*-glucoside) was de novo synthesized in all the tested TSL, while compound F13 (rutin) was exclusively found only in TSL B.

Anthocyanins. The HPLC analysis of anthocyanins resulted in the identification of cyanidin 3-*O*-glycoside (F8) and cyanidin 3-*O*-rhamnoside (F10) in all the tested shoot extracts (Table 1). The compound F10, as the dominant anthocyanin, was found in a significantly higher amount only in TSL B (1.5-fold) compared to NTS cultures. The TSL B and TSL F also displayed significantly increased F16 contents (1.6-fold) in comparison to the control shoots.

Naphthodianthrones. Transgenic and control shoots were shown to accumulate pseudohypericin (F16), as the preeminent representative of the group of naphthodianthrones, as well as hypericin (F17) and protopseudohypericin (F18) (Table 1). The TSL F showed a comparable F16 amount to the NTS cultures, while TSL H and TSL B exhibited significantly higher productivity (1.8- and 2.9-fold, respectively) compared to the control shoots. The TSL F and TSL B displayed significantly higher F17 content (1.3- and 2.3-fold, respectively) compared to the NTS. The production of F18 in all the tested TSL was markedly elevated (from 3.9- to 6.4-fold) compared to the control cultures.

Acyl-phloroglucinols. Metabolic profiling of acyl-phloroglucinols resulted in the identification of hyperforin (F19) and adhyperforin (F20) in all TSL, as well as in NTS (Table 1). In comparison to NTS, compound F19 was found in significantly higher amounts only in TSL B (1.9-fold), while the F20 content was similar in TSL B and TSL H or markedly lower in TSL F (2.1-fold). Concerning the total content of identified acyl-phloroglucinols, TSL B was identified as the best-performing clone for the accumulation of hyperforins.

Xanthones. Eleven xanthones were detected and fully identified by ESI-MS analysis in TSL and NTS cultures (Table 1). Qualitative analysis of xanthones showed the presence of six xanthones, such as mangiferin (X1), trihydroxyxanthone-sulfonate (X3), dimethylmangiferin (X4), dihydroxy-metoxyxanthone-sulfonate (X5), γ-mangostin (X9), and cadensin G (X11) in both NTS and all TSL. Two xanthones identified as brasilixanthone B (X2) and 1,3,6,7-tetrahydroxyxanthone 2-prenyl xanthone (X7) were de novo synthesized in TSL B and TSL F, while 5-*O*-methyl-2-deprenylrheediaxanthone B (X10) was found only in TSL F and TSL H. The xanthone confirmed as 1,3,6,7-tetrahydroxyxanthone 8-prenyl xanthone (X8) was found only in TSL F. Mangiferin *C*-prenyl isomer (X6) was detected in both TSL F and NTS cultures. The present results demonstrated that detected xanthones in TSL were quantified in comparable or lower amounts than those found in the NTS cultures. However, it is worth pointing out that TSL exhibited a capability for the biosynthesis of several xanthones that were not confirmed in the control shoots.

### 2.2. In Vitro Biological Activity of H. perforatum Transformed Shoots

Antidepressant activity. The MAO-A inhibitory activity of NTS and TSL expressed as IC_50_ values varied from 433.90 to 566.16 µg·mL^−1^ (Figure 1a, Table 2). However, significant differences in MAO-A inhibitory properties between NTS and TSL extracts were not noticed.

Neuroprotective activity. The outgoing results showed a great variation in the AChE inhibitory activity of shoot extracts with IC_50_ values ranging from 217.90 to 1107.23 µg·mL^−1^ (Figure 1b, Table 2). Noteworthily, all the tested TSL displayed significantly lower IC_50_ values for AChE inhibition (from 1.2- to 5.1-fold) compared to NTS. Concerning BChE, shoot extracts did not exhibit a high enough enzyme-inhibitory activity to calculate IC_50_ values, and the data were expressed as IC_25_ values ranging from 75.12 to 1257.49 µg·mL^−1^ (Figure 1c, Table 2). The TSL demonstrated exceptionally higher IC_25_ values for BChE inhibition (from 6.4- to 16.8-fold) compared to the NTS cultures. With respect to TYR, shoot extracts showed a strong capacity for enzyme inhibition, with IC_50_ values ranging from 80.66 to 150.44 µg·mL^−1^ (Figure 1d; Table 2). TSL H and TSL B displayed significantly lower IC_50_ values for TYR inhibition (1.3- and 1.9-fold, respectively) compared to the control shoots.

Antihyperglycemic activity. Shoot extracts at the tested concentration revealed an inferior capacity for α-AM inhibition and the results were expressed as IC_25_ values (Figure 1e, Table 2). The TSL and NTS cultures showed a small variation in α-AM inhibitory activity with IC_25_ values ranging from 169.91 to 277.04 µg·mL^−1^. The IC_25_ values for α-AM inhibition of TSL F and TSL H were significantly lower (1.3- and 1.6-fold, respectively) in comparison to NTS. The α-GL inhibitory activity of shoot extracts expressed as IC_50_ values ranged from 156.99 to 879.90 µg·mL^−1^ (Figure 1f, Table 2). Nevertheless, all the TSL exhibited significantly higher IC_50_ values for α-GL inhibition (from 2.2- to 5.6-fold) compared to the control shoot cultures.

Antihyperlipidemic activity. The present data demonstrate that shoot extracts had a prominent capacity for PL inhibition, with IC_50_ values ranging from 230.39 to 406.95 µg·mL^−1^ (Figure 1g, Table 2). Two clones denoted as TSL F and TSL H displayed significantly lower IC_50_ values for PL inhibition (up to 1.8-fold) in comparison to NTS. With regards to CHE, shoot samples at the tested concentration showed a high capacity for enzyme inhibition, as indicated by their IC_50_ values ranging from 102.50 to 572.04 µg·mL^−1^ (Figure 1h, Table 2). The clones TSL B and TSL H demonstrated markedly lower IC_50_ values for CHE inhibition (2.4- and 3.0-fold, respectively) compared to the control shoots. It is worth pointing out that TSL H was revealed as a stronger inhibitor of CHE with a 1.6-fold lower IC_50_ value than that noticed for simvastatin as a specific enzyme inhibitor.

### 2.3. Molecular Modelling of Phenolic Compounds from H. perforatum Transformed Shoots

The docking data for the binding energy and inhibition constant for the best four ligands (chlorogenic acid, epicatechin, pseudohypericin, and hyperforin) are shown in Table 3, while the data for the other tested ligands are presented in Appendix A.

According to the docking data for MAO-A, the best interactions with enzyme active sites were found for epicatechin (Figure 2) and cadensin G, displaying the lowest binding energy (−8.52 and −8.18 kcal·mol^−1^, respectively). The binding mode of epicatechin was established by hydrogen bonds with amino acids Tyr 197, Tyr 444, Phe 208, and Thr 336, as well as by hydrophobic interactions with amino acids Ile 335 (π-sigma), Leu 337 (π-alkyl), Tyr 407 (π-π stacked), Cys 323 (π-sulfur), and the cofactor FAD (π-π T-shaped). The interactions of cadensin G into MAO-A pocket were represented by hydrogen bonds with amino acids Phe 112, Pro 113, Asn 212, Asn 125, Thr 211 and Thr 487, as well as by four hydrophobic interactions with Ala 111 (π-alkyl) and Thr 205 (π-sigma). Other tested ligands showed a low to moderate inhibition of MAO-A, with binding energies ranging from −4.41 to −7.31 kcal·mol^−1^.

The docking results on AChE revealed that pseudohypericin and cadensin G exhibited the best docking score with binding energies of −12.00 and −10.62 kcal·mol^−1^, respectively. The best docking pose of pseudohypericin in the AChE cavity (Figure 3a,b) was stabilized by numerous hydrogen bonds with amino acids Gln 71, Asp 74, Trp 86, Asn 87, Gly 120, Tyr 124, Ser 125, Ser 203, Gly 126, Tyr 337, and His 447, as well by hydrophobic interactions with Trp 86 (π-π stacked), Tyr 124 (π-π T-shaped), Tyr 337, Phe 338, and His 447 (π-alkyl). Cadensin G-AChE complex was stabilized by hydrogen bonds with amino acids Trp 86, Gly 120, Gly 122, Ser 203, Arg 296, and Tyr 337, as well as by hydrophobic interactions with Trp 86 (π-sigma), Tyr 124, and Tyr 341 (π-π T-shaped). Other phenolic compounds showed moderate AChE inhibition with binding energies from −7.30 to −8.49 kcal·mol^−1^.

Concerning the docking data on BChE, pseudohypericin and hyperforin exhibited the best affinities to the enzyme active pocket (binding energy −14.56 and −11.06 kcal·mol^−1^, respectively). The pseudohypericin-BChE complex (Figure 3c,d) was maintained by the hydrogen bonds with amino acids Trp 82, Gly 115, Gly 121, Glu 197, Tyr 332, and His 438, hydrophobic bonds with Trp 82 (π-π stacked), Tyr 332 (π-sigma), Ala 328 and Phe 329 (π-alkyl), as well as by one electrostatic interaction with Asp 70. The best docking pose of hyperforin into the BChE active site was stabilized by one hydrogen bond to Thr 120 and multiple hydrophobic interactions with Phe 329 (π-sigma), Leu 125, Ala 328, Met 437 (alkyl), Trp 82, Trp 231, Trp 430, Phe 329, Phe 398, His 438, and Tyr 440 (π-alkyl). Other tested ligands displayed intermediate affinities towards BChE (binding energies from −5.84 to −8.94 kcal·mol^−1^).

The docking results on TYR demonstrated that chlorogenic acid had the highest affinity toward tyrosinase (Figure 4) with a binding energy of −8.09 kcal·mol^−1^, followed by pseudohypericin and hyperforin (binding energies −7.21 and −6.37 kcal·mol^−1^, respectively). Chlorogenic acid-TYR complex was maintained by the establishment of hydrogen bonds to Arg 268 and His 244, hydrophobic interaction to Phe 264 (π-π T-shaped), and coordinative bonds to Cu 400 and Cu 401. The best docking poses of pseudohypericin and hyperforin into the TYR active center were stabilized through the hydrogen bonds and hydrophobic interactions with the common amino acid residues (His 85, His 244, Gly 281, Val 283, and Pro 284). Other flavonoids and xanthones displayed low to moderate affinity towards TYR, with binding energies from −3.72 to −5.49 kcal·mol^−1^.

The molecular docking data on α-AM showed that pseudohypericin is the most active ligand (binding energy of −11.58 kcal·mol^−1^), establishing numerous interactions in the enzyme active center (Figure 5a,b). The complex of pseudohypericin with α-AM was represented through hydrogen bonds with amino acids Gly 167, Val 171, Asp 206, Asp 297, His 296, and Arg 344, hydrophobic interactions with Trp 83 (π-π T-shaped) and Leu 166 (π-alkyl), as well as electrostatic interactions with Asp 297 and Asp 340 (π-anion). The remaining phenolics displayed a moderate to strong affinity to α-AM (binding energies from −4.30 to −8.02 kcal·mol^−1^).

Concerning the docking results on α-GL, the best affinities to the enzyme active pocket were observed for pseudohypericin and hyperforin, with binding energies of −11.65 and −10.30 kcal·mol^−1^, respectively. The pseudohypericin-enzyme complex (Figure 5c,d) was maintained through hydrogen bonds to amino acids Tyr 158, Gln 279, Leu 313, Arg 315, Glu 411, and Asn 415, hydrophobic interactions to Tyr 158 (π-π T-shaped), Arg 315, and Tyr 316 (π-alkyl), as well as electrostatic interaction to Asp 242 (π-anion). The strong interactions of hyperforin with α-GL were represented by hydrogen bonds to amino acids Lys 156, Tyr 158, Asp 242, and Arg 315, as well as by numerous hydrophobic interactions to Lys 156, Tyr 158, Phe 178, Val 216, Phe 303, and Arg 315 (alkyl and π-alkyl). Intermediate affinity to α-GL was found for other phenolic compounds (binding energies from −6.33 to −8.13 kcal·mol^−1^), whereas chlorogenic acid displayed the lowest enzyme affinity (binding energy −5.09 kcal·mol^−1^).

According to the docking results, the best affinities to the PL active site were found for pseudohypericin and quercitrin (binding energies of −12.70 and −9.58 kcal·mol^−1^, respectively). The complexes of PL with pseudohypericin (Figure 6a,b) and quercitrin were represented through interactions with common amino acid residues Ser 153 and Arg 257 (hydrogen bonds), as well as Phe 78, Phe 216, Ile 79, Tyr 115, Val 260, and His 264 (hydrophobic bonding). High affinity to PL was also found for other ligands (binding energies from −7.59 to −8.76 kcal·mol^−1^), while chlorogenic acid showed the lowest docking score (binding energy of −6.14 kcal·mol^−1^).

Docking data on CHE revealed that pseudohypericin and cadensin G are the most active ligands towards the enzyme active site (binding energies −10.51 and −8.38 kcal·mol^−1^, respectively). The best docking pose of pseudohypericin into the CHE active center (Figure 6c,d) was stabilized by hydrogen bonds to amino acids Ser 194 and Met 281, as well as by multiple hydrophobic bonding with Ala 108, Met 281 (π-sigma), Met 111, Met 281, Val 272, and Val 285 (π-alkyl). The cadensin G-CHE complex was maintained through hydrogen bonds with Gly 106, Gly 107, Ala 108, and Glu 193, hydrophobic bonding with Ala 108, Met 281, Val 285 (π-alkyl), Trp 227 (π-sigma), His 435 (π-π T-shaped) and electrostatic interaction to Glu 193. Moderate inhibition to CHE was found for other tested ligands (binding energies from −4.48 to −6.28 kcal·mol^−1^).

## 3. Discussion

### 3.1. Analysis of Phenolic Compounds in H. perforatum Transformed Shoots

The present results showed that *H. perforatum* transformed shoots accumulate significant quantities of hydroxycinnamic acids. As presently established, chlorogenic acid has already been detected as a major phenolic acid in *H. perforatum* transformed shoots [23,29,30]. It is worth noting that TSL F tested here was selected as a much better source of chlorogenic acid than flowering shoots of *H. perforatum* wild-growing plants [6]. This is quite an interesting finding since chlorogenic acid attracts great interest in the pharmaceutical and nutraceutical industry as an excellent antioxidant compound with multiple biological effects [31]. The comparison of phenolic acid profiles of TSL evaluated here with those of various *H. perforatum* hairy roots-regenerated plants revealed qualitative and quantitative differences [23,30]. Previous reports indicated that *H. perforatum* transformed roots cultured on a hormone-free medium are better producers of hydroxybenzoic acids, while hairy root-regenerated shoots multiplied on the cytokinin-containing medium are characterized by the accumulation of hydroxycinnamic acids [22,29]. Also, we have recently observed that photoperiod exposition negatively affects phenolic acid production in *H. perforatum* transformed roots cultured in a liquid medium [27]. To the best of our knowledge, phenolic acid biosynthesis in *H. perforatum* in vitro cultures depends on the type of tissue culture, cell degree differentiation, solid/liquid consistency of the medium, photoperiod exposition, and phytohormone supplementation [9,10,32]. All these investigations suggested that *H. perforatum* transformed cultures could be represented as promising biotechnological systems for phenolic acid production through careful optimization of culture conditions and composition of nutrient medium.

The *H. perforatum* transformed shoots were shown to be superior producers of flavan-3-ols, exhibiting a capacity for de novo biosynthesis of oligomeric procyanidins. From our previous studies, *H. perforatum* shoots transformed with *A. rhizogenes* strain A4 accumulated only epicatechin [29], while shoots transformed with bacterial strain A4M70GUS exclusively synthesized catechin [23]. It has been shown that catechin and epicatechin are sensitive to oxidation and epimerization during the extraction procedure, depending on experimental conditions, such as the extraction time, temperature, and pH of solvents [33]. In this context, the variability observed in the detection of monomeric flavan-3-ols in *H. perforatum* transformed shoot cultures is likely to be attributed to the chemical reactions that occurred during the extraction process rather than to the effect of genetic transformation. Therefore, it is necessary to develop an appropriate method for catechins extraction under controlled epimerization and oxidation reactions to obtain relevant data for monomeric flavan-3-ols in *H. perforatum*. Noteworthily, the amounts of epicatechin and procyanidin derivatives observed here for transformed shoots were comparable to those for Hyperici herba previously reported by Tusevski et al. [6]. Taking into account that epicatechin and proanthocyanidins exhibit a plethora of biological properties [34], *H. perforatum* transformed shoots could be offered as a perspective system for the production of flavan-3-ols.

The flavonol profile in transformed shoot clones revealed the presence of quercetin derivatives that are characteristic flavonoid compounds for *H. perforatum* field-grown plants [2]. Transformed shoot extracts were enriched in quercitrin and hyperoside as dominant representatives from the group of flavonol glycosides. Both flavonols have already been shown to be preeminent quercetin glycosides in *H. perforatum* and *H. tomentosum* hairy root-regenerated shoot lines [23,29,30,35]. More importantly, transformed shoot lines showed a capability for de novo production of rutin and kaempferol 6-*C*-glucoside that were not confirmed in non-transformed shoots. From our previous studies, these flavonol glycosides have been detected in one randomly selected *H. perforatum* shoot clone transformed with *A. rhizogenes* strain A4 [29], but their presence was not confirmed in transformed plants induced by A4M70GUS-mediated transformation [23]. In addition, *H. perforatum* plant lines transformed with ATCC 15,834 have been shown to accumulate rutin, while the presence of kaempferol derivatives has not been confirmed [30]. These findings indicate that the production of quercetin and kaempferol derivatives in *H. perforatum* transformed shoots might be dependent on the *A. rhizogenes* strain used for the transformation process. Previously, rutin-free chemotypes have been reported for wild Italian *Hypericum* species, suggesting that *H. perforatum* is characterized by a high degree of chemical polymorphism [36]. Furthermore, rutin has been identified only in the *veronense* subspecies, but it has been completely absent in the *perforatum* and *angustifolium* subspecies [37]. Thus, the variations of *H. perforatum* transformed plants for rutin production could also be attributed to the plant chemotypes/subspecies used for transformation with *A. rhizogenes*. Regarding the flavonol aglycones, shoot clones were shown to be effective producers of quercetin, which is in line with previous reports for *Hypericum*-transformed plants [23,30,35].

Despite the great potential of *H. perforatum* transformed shoots to accumulate various groups of flavonoids, there is still a lack of data on the production of anthocyanins. This study revealed for the first time the accumulation of anthocyanin glycosides in *H. perforatum* shoot lines, such as cyanidin 3-*O*-glucoside and cyanidin 3-*O*-rhamnoside. Both anthocyanins have previously been identified in *H. perforatum* callus and shoot cultures [11]. These authors suggested that tissue differentiation and shoot development from unorganized *Hypericum* cultures were imperative processes for the biosynthesis of cyanidin glycosides. In this context, the superior accumulation of anthocyanins in TSL B reported here could be related to its fast-growing rate and high biomass yield [28]. Thus, the establishment of *H. perforatum* transformed shoots in an advanced stage of differentiation could be proposed as an alternative system for anthocyanins production.

The presence of hyperforin and adhyperforin as major bioactive acyl-phloroglucinols was confirmed in *H. perforatum* transformed shoots. Even hyperforins have already been identified in liquid-cultured *H. perforatum* transformed shoots [30], the shoot lines analyzed here were found to be better producers of hyperforin and adhyperforin. It has been previously established that hyperforin production in *H. perforatum* shoots is related to tissue differentiation and leaf development [10], the number and size of translucent glands as the main accumulation sites [38], or the presence of cytokinins in the culture medium [39]. These observations indicate that acyl-phloroglucinol production in transformed shoot lines is attributed to their high multiplication and/or differentiation rate on solid medium supplemented with benzyladenine. Even though we did not evaluate the number or size of translucent glands on the leaves of *H. perforatum* shoot lines, the capacity of these glandular structures for acyl-phloroglucinols accumulation might be affected by *A. rhizogenes*-mediated transformation. The immense potential of TSL B for hyperforin accumulation could also be related to the structure and/or metabolic activity of translucent glands on the leaves. Thus, the selection of high-producing *H. perforatum* shoot lines could have great importance for further obtainment of standardized extracts enriched in hyperforins.

Metabolic profiling of *H. perforatum* shoot extracts demonstrated that xanthones represent the major class of phenolic compounds. In addition, the transformed shoots showed a capability for significant production of mangiferin, trihydroxyxanthone-sulfonate, dihydroxy-metoxyxanthone-sulfonate, and cadensin G, as well as de novo biosynthesis of brasilixanthone B, 5-*O*-methyl-2-deprenylrheediaxanthone B, 1,3,6,7-tetrahydroxyxanthone 2-prenyl xanthone and 1,3,6,7-tetrahydroxyxanthone 8-prenyl xanthone. The capability of *H. perforatum* shoot clones for immense accumulation of xanthones represented a potentially interesting finding since it has been confirmed that these metabolites are accumulated in root exodermis and endodermis, playing a substantial role in the defense against pathogens [40]. The defensive role of xanthones has also been confirmed through the elicitation of *H. perforatum* root cultures with chitosan that simulates fungal pathogen attack [41]. Additionally, *H. perforatum* cell suspensions co-cultured and elicited with *Agrobacterium* have been shown to synthesize considerable quantities of xanthones, highlighting their antioxidant and antibacterial effects [20]. Considering all these data, it could be hypothesized that the accumulation of xanthones in *H. perforatum* hairy root-regenerated shoots is related to the stress-induced defense response upon the *A. rhizogenes* transformation process. Further investigation of xanthone biosynthesis regulation in *H. perforatum* transformed shoots would be of particular interest for the large-scale production of these bioactive compounds for medicinal purposes.

### 3.2. Biological Activity of H. perforatum Transformed Shoots

Depression is a common mental disorder that is characterized by the deficiency of amine neurotransmitters in the brain. The antidepressant activity of natural and synthetic compounds has been represented by the inhibition of MAO-A, which prevents the catabolism of neurotransmitters leading to enhanced monoaminergic activity [42]. In this study, MAO-A inhibitory activity was not significantly changed among different *H. perforatum* transformed clones, including non-transformed shoots. Docking data showed that epicatechin and cadensin G are the most active MAO-A inhibitors exhibiting K_i_ values in nanomolar levels. In this line, several studies have reported that flavonoid aglycones and xanthones from *Hypericum* field-grown plants represent efficient MAO-A inhibitory compounds [6,42]. The strongest docking score of epicatechin into the MAO-A active site represents an unusual outcome since it has been established that the absence of a double bond at the C2 and C3 position and non-planar conformation of epicatechin is not a favorable structure for MAO-A inhibition [43]. However, it has been shown that glycosylation with sugar moiety significantly decreases the MAO inhibitory potential of flavonoid compounds [42]. Taking this into consideration, the strong MAO-A inhibitory effect of epicatechin could be ascribed to its aglyconic nature, since glycosylated flavonoids do not fit into the MAO pocket due to the steric hindrance [22]. The mood-modulating effect of epicatechin has also been confirmed in vivo in mice by enhancing the production of neurotransmitters in the brain, a decrease in MAO-A expression, and subsequent inhibition of monoamines enzymatic degradation [44]. For xanthones, it has been presented that the number and position of OH and OCH_3_ substitutions could markedly affect MAO-A inhibitory activity [45]. Our recent docking data indicated that paxanthone and other dihydroxyxanthone derivatives represent the best MAO-A inhibitors that could be responsible for the in vitro antidepressant activity of *H. perforatum* transformed root extracts [27]. Even though the MAO-A inhibitory properties of cadensin G have not been evaluated yet, further structure–activity relationship studies of xanthones from *H. perforatum* would be of fundamental importance in medicinal chemistry for the design of novel compounds to improve human mood disorders.

Alzheimer’s disease is one of the most common progressive neurodegenerative disorders related to a deficiency of acetylcholine and butyrylcholine, which are hydrolyzed by AChE and BChE, respectively [46]. Parkinson’s disease is a neurological disorder resulting from dopamine shortage in the brain. The TYR enzyme catalyzes the conversion of L-tyrosine into dopamine and its excessive activity causes dopamine neurotoxicity associated with Parkinson’s disease [47]. Thus, the inhibition of AChE, BChE, and TYR is considered an important strategy in the management of neurodegenerative diseases.

The outgoing results demonstrated that shoot extracts have a markedly higher capability for the inhibition of AChE in comparison to BChE. In comparison to non-transformed shoots, transformed shoots showed a superior capacity for AChE inhibition, but inferior BChE inhibitory properties. Despite the similar properties of AChE and BChE, these enzymes exhibited distinct substrate activity, specificity, and kinetics [48]. Considering the complex composition of tested shoot extracts, the present findings suggested that the AChE and BChE inhibitory activity of *H. perforatum* shoots could be explained by the individual effects of distinct phenolic compounds with specific structural characteristics. According to the docking data, pseudohypericin, hyperforin, and cadensin G were presented as remarkable inhibitors of cholinesterase enzymes, displaying the lowest binding energies. Our recent in silico studies suggested that the AChE and BChE inhibitory effects of *H. perforatum* transformed roots were related to their strong capacity for xanthone biosynthesis [22,27]. It has been noted that variations in the hydroxylation and methoxylation pattern of the xanthone nucleus could result in selective inhibition of AChE and BChE, while some structural moieties have great importance for dual cholinesterase inhibition [49]. Since transformed shoots accumulated a plethora of xanthones, the association of particular xanthone derivatives from *H. perforatum* with cholinesterase inhibitory activity still represents a major challenge. Further in vivo and clinical validation studies would be of crucial significance to confirm the efficacy of xanthones as anti-Alzheimer’s drugs.

Some studies have reported that *Hypericum* spp. could be efficiently used for the treatment of Alzheimer’s disease due to their cholinesterase inhibitory properties originating from hypericins and hyperforins [50,51,52]. Our molecular docking data revealed that pseudohypericin binds much more tightly with AChE and BChE than hyperforin and cadensin G by establishing H-bonds and hydrophobic interactions with several amino acid residues of both enzymes. Additionally, the K_i_ values of pseudohypericin were 1.59 nM and 259.09 pM for AChE and BChE, respectively, while those values for hyperforin and cadensin G varied in nanomolar levels (7.75 nM–860.00 nM). In agreement with these findings, Thakur et al. [53] have screened 85 herbal compounds through an in silico docking study against human AChE and revealed that hypericin represents the most promising cholinesterase inhibitor. Hypericin and pseudohypericin have a similar structure and pseudohypericin possesses a more hydrophilic structure due to the presence of a CH_2_OH instead of a CH_3_ group in hypericin. This additional OH group in the structure of pseudohypericin potentially makes this naphthodianthrone a better cholinesterase inhibitor compared to hypericin, because these functional groups are responsible for the establishment of strong H-bonds in the enzymatic pocket. The docking data also selected hyperforin as a potent neuroprotective compound with stronger inhibitory activity against BChE compared to AChE. In accordance with this, Orhan et al. [52] have shown that hyperforin is ineffective against AChE, but a highly active BChE inhibitor since this phloglucinol derivative blocks the normal functioning of enzyme and complements some hydrophobic residues of the enzyme cavity. Even though there are not as many studies on BChE inhibition as for AChE and the relationship between inhibitor structures and their activity is scarce, present in silico data indicate that hydrophobic interactions involving prenyl side chains of hyperforin are the main prerequisites for powerful BChE inhibitory activity. Taken together, these data implied that the overall neuroprotective activity of *H. perforatum* transformed shoot extracts could be the result of the cumulative effect of complex mixtures of phenolics, wherein hypericins, hyperforins, and xanthones are likely to have a pivotal role.

The present results showed that transformed shoots have a considerable capacity for TYR inhibitory activity compared to non-transformed cultures. The computational docking analysis revealed that chlorogenic acid is the most efficient inhibitor of TYR, followed by pseudohypericin and hyperforin. However, the K_i_ values of those ligands for TYR did not show a marked variation (from 1.17 µM to 21.33 µM). In agreement with these data, previous in vitro and in silico studies confirm the relationship between TYR inhibitory properties and chlorogenic acid content in the aerial parts of wild-growing *H. perforatum* [6] and other *Hypericum* species [54]. Even though several studies suggested that *Hypericum* spp. represented important sources of metabolites with TYR inhibitory capacity [5,47], their contribution to the bioactivity has not been determined yet. However, it is interesting to note that TSL B, with the best TYR inhibitory property, accumulated an exceptionally low amount of chlorogenic acid, suggesting that hypericins and hyperforins as preeminent compounds in this clone may be responsible for enzyme inactivation. As presently established, the docked poses of pseudohypericin and hyperforin in the TYR pocket were mainly stabilized through hydrophobic interactions with the enzyme amino acids. Our findings suggested that hydrophobic interactions involving the polycyclic moieties from hypericins and prenyl groups of hyperforins might be responsible for the TYR inhibitory property of transformed shoots.

Diabetes is a chronic disease characterized by hyperglycemia and an alteration in carbohydrates, fats, and protein metabolism due to a shortage in the secretion of insulin or its impaired action. The carbohydrate-hydrolyzing enzymes α-AM and α-GL play a major role in the degradation of starch and oligosaccharides into monosaccharides that can be absorbed in the digestive tract. One therapeutic approach for reducing post-prandial hyperglycaemia is the prevention of glucose absorption through the inhibition of α-AM and α-GL [55]. The data for in vitro antihyperglycemic activity showed that shoot extracts are better inhibitors of α-GL than α-AM. Regarding transformed shoot clones, only TSL F and TSL H displayed significantly higher α-AM inhibitory activity compared to non-transformed shoots. In contrast, all the transformed clones exhibited a markedly lower production of α-GL inhibitory compounds. It has been reported that a combination of catechin and acarbose at low concentrations synergistically affected α-GL inhibition, while this effect turned out to be antagonistic at high concentrations [56]. This could be confirmed by the evidence that TSL F enriched in flavan-3-ols and xanthones exhibited the lowest α-GL inhibition values. Thus, we hypothesized that the abundance of epicatechin and xanthones in *H. perforatum* transformed cultures exerted an antagonistic effect on α-GL inhibitory activity.

Molecular docking analyses demonstrated that pseudohypericin and hyperforin are the most effective inhibitors of α-AM and α-GL, implying that these major phenolics from *H. perforatum* shoot extracts are involved in enzyme inhibition. Although several in vitro and in silico studies have proposed that various components from *Hypericum* sp. possess α-AM and α-GL inhibitory activities [22,27,54], this study provides the first information indicating that pseudohypericin and hyperforin could inhibit the activity of carbohydrate-hydrolyzing enzymes. Our docking experiments suggest that H-bonds between phenolic OH groups of pseudohypericin and amino acid residues of both α-AM and α-GL are the main force for efficient enzyme inactivation. Also, it has been noted that H-bonding might increase the hydrophobicity of α-GL, which was found to be essential for improving the stability of the enzyme–ligand complex [57]. As presently established, the main amino acid residues around acarbose as a specific α-GL inhibitor were recorded to include Tyr 158, Ser 240, Asp 242, Gln 279, Arg 315, Tyr 316, and Glu 411 [58]. These data indicated that naphthodianthrones and acarbose bind to α-GL at the common amino acid residues and the inhibition mechanism of pseudohypericin could be similar to that of the specific enzyme inhibitor. Additionally, Dong et al. [57] have demonstrated that hypericin represents a strong, reversible, and competitive α-GL inhibitor through the establishment of multiple H-bonds in the enzyme active site. According to our docking data, the methyl group of pseudohypericin was shown to interact with Arg 315 and Tyr 316 at the hydrophobic region of α-GL, which was known to be located at the entrance of the enzyme active site [57]. Yamamoto et al. [59] have demonstrated that the active pocket of α-GL (isomaltase from *Saccharomyces cerevisiae* in complex with maltose) is shallower than that of other oligo-1,6-glucosidases and its entrance to the active site is narrowed by Tyr 158 and His 280, as well as a loop 310–315. The present docking data revealed that pseudohypericin has the capacity to establish hydrogen bonds to Leu 313 and Arg 315 from this loop, which act as a gateway for substrate binding, and might be responsible for a “trap-release” mechanism of substrate hydrolysis [59]. Therefore, it could be assumed that the binding of pseudohypericin to these gate keeper amino acid residues from the active site blocked the entry of the substrate, leading to inhibition of α-GL activity. With respect to α-AM, pseudohypericin with the best K_i_ value of 3.27 nM established hydrogen bonds to Asp 206 as the catalytic nucleophile and conserved Asp 297, which may play a key role in substrate binding [60]. It has been reported that the catalytic triad of *Aspergillus oryzae a*-AM enzyme responsible for the cleavage of glycosidic bonds consisted of Glu 230, Asp 206, and Asp 297 [61]. These results indicated that the low binding energy, the capability to interact with key amino acid residues of glycosidic cleavage (Asp 206 and Asp 297), as well as the large number of interactions in the active pocket of *a*-AM can explain the strong inhibitory activity by pseudohypericin.

This study also reports that hyperforin as lipophilic polyprenylated acylphloroglucinol from *H. perforatum* represents a strong inhibitor of α-AM and α-GL, with K_i_ values of 1.33 µM and 28.32 µM, respectively. According to the molecular modeling analysis, the docked pose of hyperforin into α-AM and α-GL pocket was stabilized mainly through hydrophobic interactions that involve prenyl groups, while only a few H-bonds were established. Our previous reports showed that prenylated xanthones from wild-growing roots and in vitro transformed roots of *H. perforatum* could contribute to α-AM and α-GL inhibitory activities [6,22]. These findings are in good agreement with previous reports indicating that prenylation patterns of various compounds might significantly influence α-AM and α-GL activity [62,63]. More importantly, we have recently confirmed that *H. perforatum* extracts rich in hyperforin exhibit a strong insulinotropic effect in streptozotocin-induced diabetic rats [1]. As presently established from the docking data, hypericin and hyperforin, as major components of *H. perforatum*, appear to act in a complementary manner in strengthening the antidiabetic efficacy through α-AM and α-GL inhibitory activity. Further studies should be focused on the evaluation of the molecular mechanism by which hypericins and hyperforins from *H. perforatum* transformed shoots exhibit antihyperglycemic properties and regulate carbohydrate metabolism in diabetic laboratory animals.

Hyperlipidemia represents a metabolic disorder which is characterized by increased levels of triglycerides and cholesterol in the blood. The PL has a pivotal role in the enzymatic hydrolysis of triglycerides to diacylglycerols and fatty acids, while CHE esterase catalyzes the hydrolysis of cholesterol esters to free cholesterol [64]. These enzyme products form mixed micelles which can be absorbed from the enterocytes into blood circulation. One important strategy for the prevention and treatment of hyperlipidemia includes delayed digestion of dietary triglycerides and cholesterol esters through inhibition of PL and CHE in the small intestine.

It has been shown that *H. perforatum* possesses beneficial effects for the treatment of hyperlipidemia-related metabolic syndrome [25,65], but the contribution of phytochemicals from this plant to the antihyperlipidemic activity is scarce. Our data showed that transformed shoots with an increased accumulation of naphthodianthrones are more powerful inhibitors of PL compared to non-transformed shoots. In this view, pseudohypericin was selected as the most efficient ligand molecule for PL inhibition, exhibiting a K_i_ value in the picomolar level. Accordingly, we have recently confirmed the relationship between pseudohypericin content and the PL inhibitory effect of photoperiod-exposed transformed roots of *H. perforatum* [27]. Moreover, previous in vitro and in silico results revealed that hypericin and pseudohypericin are excellent PL inhibitors, indicating that these naphthodianthrones are major contributors to the antiobesity effects of *H. perforatum* [66]. This finding was corroborated by the present in silico analysis, reporting that pseudohypericin with an abundance of benzene rings and oxygen atoms could be well docked into the catalytic pocket of PL through H-bonding and multiple hydrophobic π-π interactions.

One of the main accomplishments of this study was the comparable or even stronger CHE inhibitory activity of transformed clones (TSL B and TSL H) than simvastatin as a specific enzyme inhibitor. The in silico study indicated that pseudohypericin fits well in the CHE binding pocket, exhibiting the best docking score (−10.51 kcal·mol^−1^) and K_i_ value (19.91 nM), thus potentially preventing the participation of enzymes in ester hydrolysis. This outstanding inhibitory effect of pseudohypericin against CHE could be related to the establishment of H-bonding to Ser 194 as the enzyme active center, as well to other interactions with neighboring amino acids (Gly 107, Ala 108, Ala 195, and His 435). In this view, hypericin derivatives isolated from *H. perforatum* have been shown to improve the disturbance in lipid metabolism and hypercholesterolemia in obese mice [65]. Taking into account that this is the first docking report for strong CHE inhibitory activity of pseudohypericin, future investigations should be directed at the isolation of naphthodianthrones from *H. perforatum* and the evaluation of their antihypercholesterolemic effects.

The docking data also demonstrated that cadensin G is the second most powerful CHE inhibitor (K_i_ value of 723.46 nM), which is in agreement with previous in vitro and in silico data assuming that xanthones are major anti-CHE compounds in *H. perforatum* transformed roots [27]. Additionally, treatments of diabetic animals with *H. perforatum* transformed root extracts have been shown to improve dysregulated lipid metabolism, such as serum levels of cholesterol and triacylglycerols [25]. Accordingly, it looks appropriate to attribute the CHE inhibitory effect of *H. perforatum* transformed shoots to different extract components, which may include synergistic relationships.

## 4. Materials and Methods

### 4.1. Plant Material and Culture Conditions

The HR cultures of *H. perforatum* were obtained by genetic transformation with *A. rhizogenes* strain A4 through the successful PCR amplification of *rolB* gene [18]. Each individual root regenerating from an infected explant was regarded as an HR clone arising from a single transformation event, which is distinct from the other due to different integration sites and copy numbers of Ri T-DNA genes. In this context, we have previously established fifteen different HR clones (HR A-HR O) that were subcultured monthly and maintained in Petri dishes on a hormone-free MS/B5 solid medium under darkness [24]. Root segments (2–3 cm) isolated from three independent HR clones denoted as HR B, HR F, and HR H were separately inoculated into 350 mL glass jars with 70 mL solid MS/B5 hormone-free medium and exposed to 16 h of photoperiod (light intensity of 50 μmol·m^2^·s^−1^). Along with HR cultures, root segments from in vitro-grown seedlings designated as non-transformed roots (NTR) were also exposed to the same photoperiodic conditions. After 3–4 weeks of cultivation, light-exposed HR B, HR F, and HR H were spontaneously regenerated into distinct transformed shoot lines (TSL B, TSL F, and TSL H) corresponding to those HR clones from which they originated. Also, NTR were regenerated into non-transformed shoots (NTS) under the same conditions. The apical parts of TSL and NTS with 2–3 pairs of leaves were isolated and inoculated into solid MS/B5 medium supplemented with 0.2 mg·L^−1^ N^6^-benzylaminopurine for subsequent shoot proliferation. The multiplied shoot cultures were maintained in a culture room at 25 ± 1 °C, 50–60% relative humidity, and the abovementioned lighting conditions. After one month of cultivation, the collected shoot biomass was dried under reduced pressure and used for phytochemical characterization and enzyme-inhibitory activities.

### 4.2. Chromatographic Identification and Quantification of Phenolic Compounds

Powdered shoot samples were homogenized with 80% methanol in an ultrasonic bath at 4 °C for 20 min and methanolic homogenates were centrifuged at 11,000 rpm for 15 min. The supernatants were filtered through 0.2 μm filters and used for the identification and quantification of phenolics by HPLC analysis. Phenolic compounds in shoot samples were evaluated by HPLC/DAD/ESI-MS^n^ analysis on an Agilent 1100 system (Agilent Technologies, Waldbronn, Germany) with a diode array detector coupled with an ion trap mass detector and controlled by ChemStation (Agilent, v.08.03) and LCMSD (Agilent, v.6.1.) software. Despite the phenolic profile analyzed by HPLC analysis, we also presented here our previously published data concerning the naphthodianthrone contents of tested TSL assayed by the ACQUITY ultra-performance liquid chromatography (UPLC) system (H-class Waters, Milford, MA, USA) with a dual-wavelength tunable UV/Vis (TUV) detector controlled by ACQUITY UPLC Console and MassLynx v4.1 software [28]. The comprehensive protocols for the separation of compounds, mobile-phase composition, and gradient programs were described in our recent studies [22,27,28]. The chromatograms were read at 260 nm for acyl-phloroglucinols and xanthones, 280 nm for flavanols, 330 nm for phenolic acids, 350 nm for flavonols, 520 nm for anthocyanins, and 590 nm for naphthodianthrones. The commercial standards of chlorogenic acid, (epi)catechin, quercetin, cyanidin 3-*O*-glucoside chloride, hypericin, pseudohypericin, hyperforin, and mangiferin were used as reference compounds. The identification of compounds was performed by the UV/Vis, MS, and MS^2^ spectra, retention times of the standards (Appendix A), and previously published data [18,22,23,27,28,29]. Quantification of the identified compounds was made according to the area under the peaks in UV chromatogram. Each sample was analyzed in triplicate, and the relative standard deviation of the phenolic content ranged from 0.2 to 5%.

### 4.3. In Vitro Biological Activities

Shoot methanolic extracts were evaporated in a freeze-dryer (Labconco, Kansas City, MO, USA), and dry extracts were dissolved in 50% dimethyl sulfoxide. The stock solutions of TSL and NTS extracts were serially diluted for performing enzyme-inhibitory assays. The protocols for MAO-A, AChE, BChE, TYR, α-AM, α-GL, PL, and CHE inhibitory activities were presented in our previous studies [6,22,27]. The concentration of shoot extracts that provide 50% enzyme inhibition (IC_50_) was determined by GraphPad Prism v.8.0 software (GraphPad Software Inc., San Diego, CA, USA). Three replicates were used for all enzyme-inhibitory tests.

### 4.4. Molecular Modelling

The structures of the enzymes used in this study were downloaded from the Protein Data Bank RSCB PDB: human MAO-A (PDB ID: 2Z5X), AChE from *Electrophorus electricus* (PDB ID: 1C2B), human BChE (PDB ID: 4BDS), TYR from *Agaricus bisporus* (PDB ID: 2Y9X), α-AM from *Aspergillus oryzae* (PDB ID: 7P4W), isomaltase from *Saccharomyces cerevisiae* (PDB ID: 3AXI), triacylglycerol lipase/colipase complex from *Sus scrofa* (PDB ID: 1ETH), and human bile salt activated lipase (PDB ID: 1F6W). The crystal structures of the enzymes were prepared by AutoDock Tools 4.2 as reported in our previous studies [22,27]. According to the quantification data on shoot extracts, chlorogenic acid, epicatechin, quercitrin, hyperoside, cyanidin 3-*O*-rhamnoside, pseudohypericin, hyperforin, and cadensin G were found to be dominant compounds and used as phytoligands for the docking analysis (Appendix A). The phytoligand molecules were downloaded from PubChem online database [67] or sketched in ACD/ChemSketch v.2021.1.3 software and then subjected to automatic 3D Structure Optimization (2018.2.1). The atomic charge and potential of the ligands were computed using VEGA ZZ program (3.1.2) using TRIPOS force field along with Gasteiger charges [68]. After this optimization procedure, the ligand structures were saved in pdbqt format by AutoDock Tools 4.2. The AutoDock 4.2 software was used for resolving the interactions between phytoligands and enzyme receptors using the Lamarckian Genetic Algorithm [69]. Standard docking protocol for rigid protein and flexible ligands was performed with 10 independent runs per phytoligand. The best ligand-binding conformation was selected based on the lowest binding energy and inhibition constant, as well as the type of interaction and intermolecular distance between the ligand atoms and enzyme amino acid residues. The docking results were analyzed by Discovery Studio Visualizer 16.1 (Accelrys, San Diego, CA, USA).

### 4.5. Statistical Analyses

The quantitative data for the identified phytochemicals and enzyme-inhibitory activities in the shoot samples were expressed as mean values with standard deviation. The statistical analyses were performed using the software program STATISTICA for Windows (v. 8.0; StatSoft Inc., Tulsa, OK, USA). The mean values were compared by one-way ANOVA and the significant differences (*p* < 0.05) were post-hoc evaluated using Duncan’s multiple range test.

## 5. Conclusions

This study presents a comparative evaluation of the phytochemical profile and in vitro biological activities of transformed shoot lines and non-transformed shoots of *Hypericum perforatum*. Transformed shoots were observed to be a rich source of hydroxycinnamic acids, flavan-3-ols, flavonols, anthocyanins, naphthodianthrones, acyl-phloroglucinols, and xanthones. In comparison to non-transformed shoots, transformed shoot lines exhibited a higher capability for the inhibition of enzymes related to neurodegeneration (acetylcholinesterase and tyrosinase), diabetes (α-amylase), and obesity (pancreatic lipase and cholesterol esterase). The computational docking analysis revealed that hypericins, hyperforins, cadensin G, epicatechin, and chlorogenic acid were the main contributors to the neuroprotective, antidiabetic, and antiobesity activity of transformed shoot extracts. These findings represent a starting point for further isolation of phenolic compounds from *H. perforatum* transformed shoots that might be responsible for various biological activities.

## Figures and Tables

**Figure 1 molecules-29-03893-f001:**
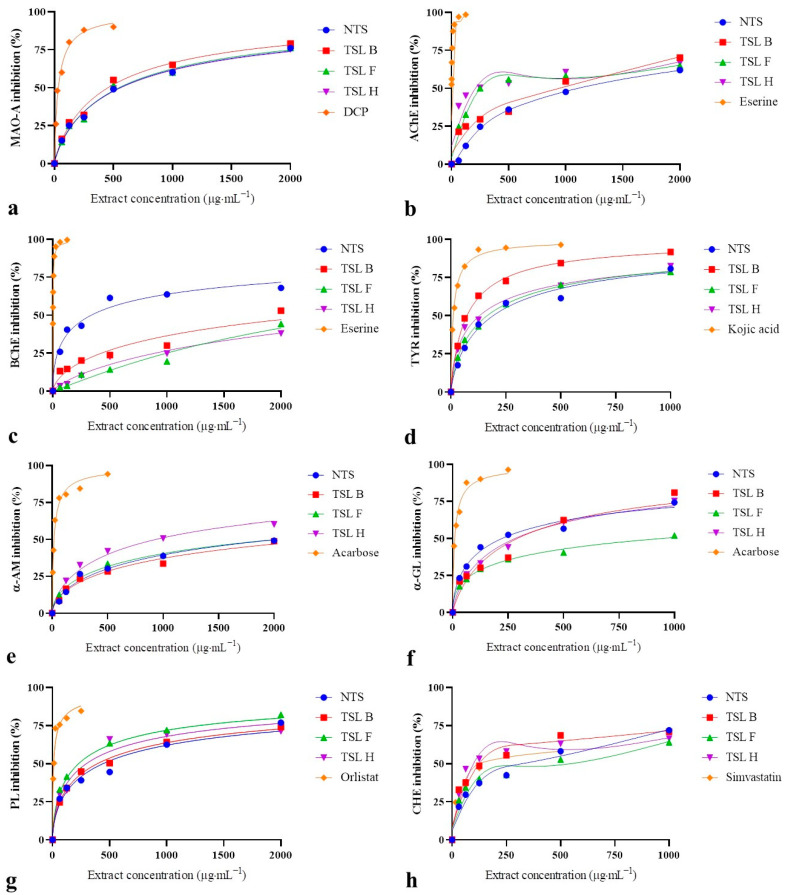
Inhibitory activity (%) of *Hypericum perforatum* transformed shoot extracts against (**a**) momoamine oxidase-A (MAO-A), (**b**) acetylcholinesterase (AChE), (**c**) butyrylcholinesterase (BChE), (**d**) tyrosinase (TYR), (**e**) α-amylase (α-AM), (**f**) α-glucosidase (α-GL), (**g**) pancreatic lipase (PL) and (**h**) cholesterol esterase (CHE). NTS: non-transformed shoots, TSL B, TSL F and TSL H: transformed shoot lines, DCP: 2,4-dichlorophenol.

**Figure 2 molecules-29-03893-f002:**
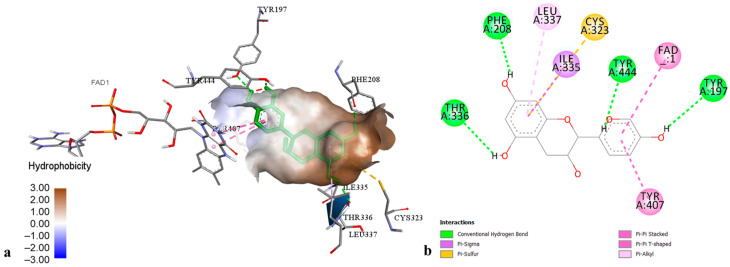
The best-ranked docking pose (**a**) and key interactions (**b**) of epicatechin in the active site of monoamine oxidase-A.

**Figure 3 molecules-29-03893-f003:**
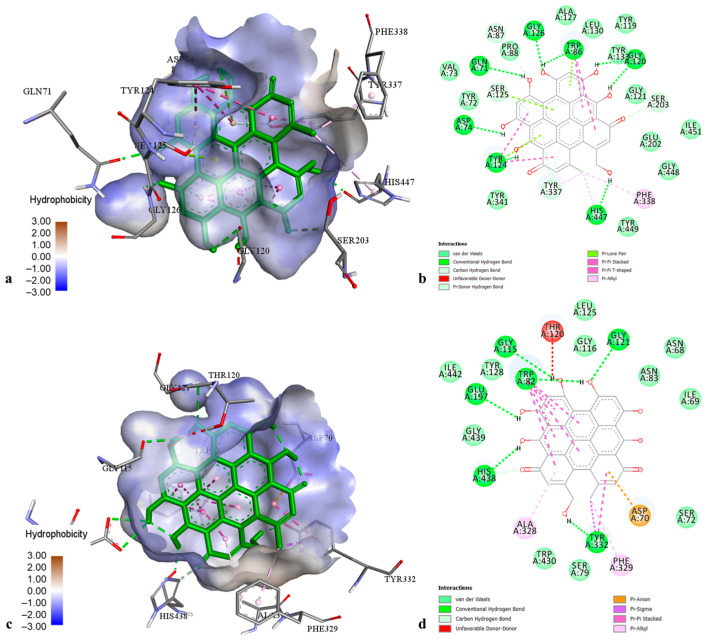
The best-ranked docking pose (**a**) and key interactions (**b**) of pseudohypericin in the active site of acetylcholinesterase. The best-ranked docking pose (**c**) and key interactions (**d**) of pseudohypericin in the active site of butyrylcholinesterase.

**Figure 4 molecules-29-03893-f004:**
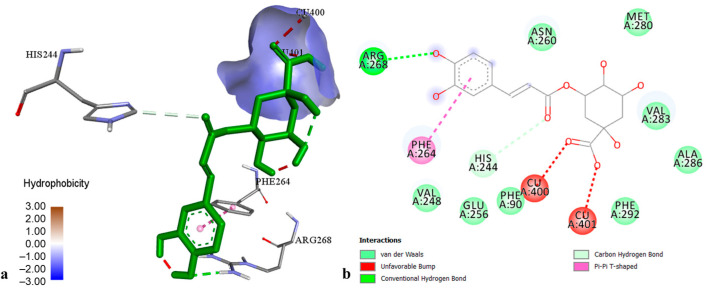
The best-ranked docking pose (**a**) and key interactions (**b**) of chlorogenic acid in the active site of tyrosinase.

**Figure 5 molecules-29-03893-f005:**
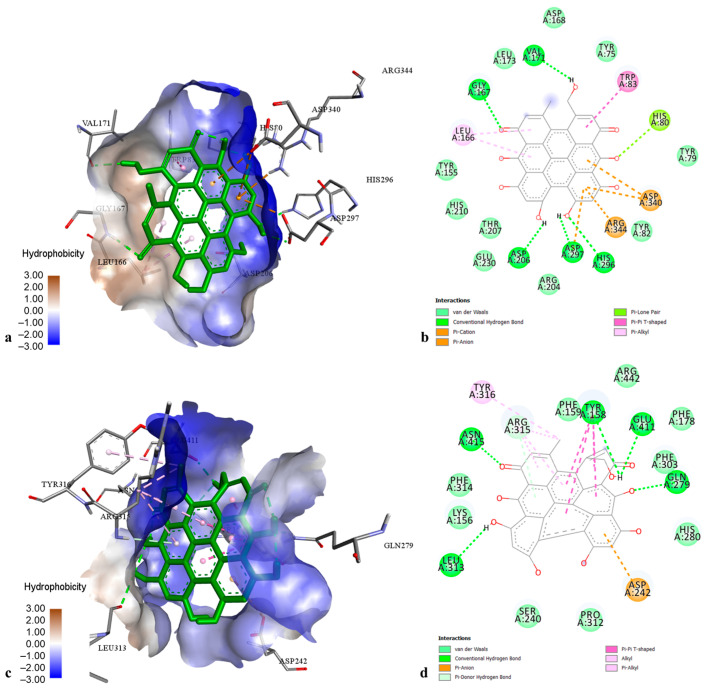
The best-ranked docking pose (**a**) and key interactions (**b**) of pseudohypericin in the active site of α-amylase. The best-ranked docking pose (**c**) and key interactions (**d**) of pseudohypericin in the active site of α-glucosidase.

**Figure 6 molecules-29-03893-f006:**
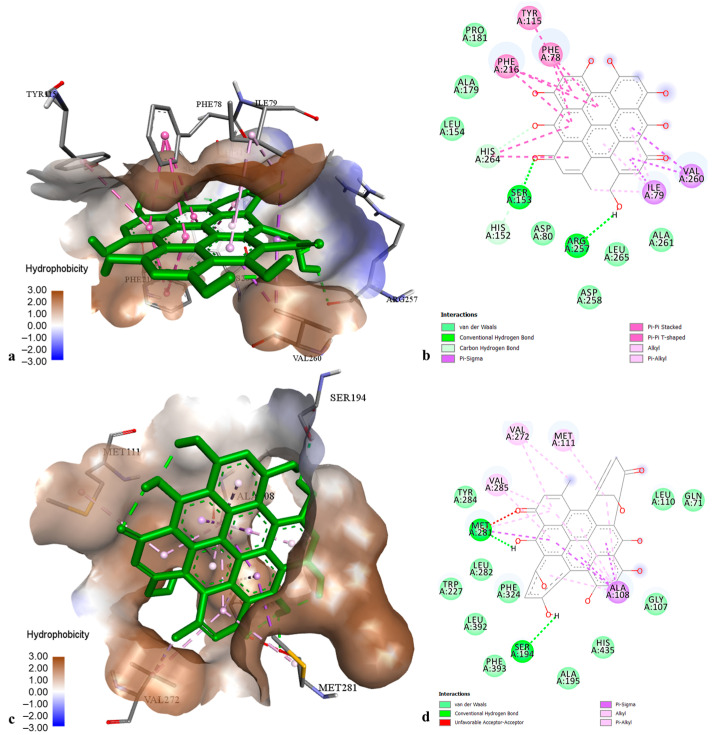
The best-ranked docking pose (**a**) and key interactions (**b**) of pseudohypericin in the active site of lipase. The best-ranked docking pose (**c**) and key interactions (**d**) of pseudohypericin in the active site of cholesterol esterase.

**Table 1 molecules-29-03893-t001:** Phenolic compound contents in *Hypericum perforatum* shoot cultures *.

Peak	Phenolic Compounds	NTS	TSL B	TSL F	TSL H
Phenolic acids					
F2	Chlorogenic acid	8.75 ± 1.01 ^c^	0.77 ± 0.06 ^a^	12.44 ± 1.63 ^d^	4.04 ± 0.89 ^b^
F3	3-*p*-Coumaroylquinic acid	1.97 ± 0.43 ^b^	0.46 ± 0.27 ^a^	0.21 ± 0.16 ^a^	0.97 ± 0.52 ^ab^
F5	3-Feruloylquinic acid	0.86 ± 0.07 ^b^	n.d.	0.29 ± 0.02 ^a^	1.23 ± 0.16 ^c^
Flavan-3-ols					
F1	(epi)catechin-(epi)gallocatechin dimer	n.d.	n.d.	4.17 ± 0.53	n.d.
F4	Procyanidin B2	n.d.	1.54 ± 0.17 ^a^	4.14 ± 0.38 ^b^	1.40 ± 0.11 ^a^
F6	Procyanidin trimer	n.d.	n.d.	3.71 ± 0.55	n.d.
F7	(epi)catechin	5.64 ± 0.41 ^b^	4.97 ± 0.25 ^ab^	6.05 ± 0.82 ^b^	4.62 ± 0.17 ^a^
Flavonol glycosides and aglycons					
F9	Quercetin 6-*C*-glucoside	0.91 ± 0.05 ^a^	n.d.	0.95 ± 0.09 ^a^	n.d.
F11	Kaempferol 6-*C*-glucoside	n.d.	0.81 ± 0.04 ^b^	0.46 ± 0.07 ^a^	0.97 ± 0.11 ^b^
F12	Hyperoside (quercetin 3-*O*-galactoside)	2.22 ± 0.20 ^b^	1.47 ± 0.23 ^a^	2.41 ± 0.15 ^b^	1.66 ± 0.09 ^a^
F13	Rutin (quercetin 3-*O*-rutinoside)	n.d.	0.45 ± 0.02	n.d.	n.d.
F14	Quercitrin (quercetin 3-*O*-rhamnoside)	3.89 ± 0.52 ^ab^	4.15 ± 0.39 ^b^	3.06 ± 0.67 ^a^	3.00 ± 0.23 ^a^
F15	Quercetin	0.31 ± 0.02 ^a^	0.53 ± 0.07 ^b^	0.44 ± 0.05 ^b^	0.42 ± 0.05 ^b^
Anthocyanins					
F8	Cyanidin 3-*O*-glycoside	0.08 ± 0.01 ^a^	0.13 ± 0.02 ^b^	0.13 ± 0.01 ^b^	0.11 ± 0.02 ^ab^
F10	Cyanidin 3-*O*-rhamnoside	1.42 ± 0.15 ^ab^	2.08 ± 0.13 ^c^	1.25 ± 0.10 ^a^	1.75 ± 0.21 ^bc^
Naphthodianthrones					
F16	Pseudohypericin	0.66 ± 0.03 ^a^	1.91 ± 0.06 ^c^	0.76 ± 0.06 ^a^	1.18 ± 0.06 ^b^
F17	Hypericin	0.03 ± 0.00 ^b^	0.07 ± 0.00 ^d^	0.02 ± 0.00 ^a^	0.04 ± 0.00 ^c^
F18	Protopseudohypericin	0.09 ± 0.01 ^a^	0.48 ± 0.03 ^c^	0.35 ± 0.03 ^b^	0.58 ± 0.02 ^d^
Acyl-phloroglucinols					
F19	Hyperforin	1.70 ± 0.13 ^b^	3.31 ± 0.27 ^c^	1.60 ± 0.10 ^b^	1.07 ± 0.08 ^a^
F20	Adhyperforin	0.40 ± 0.03 ^b^	0.35 ± 0.03 ^b^	0.19 ± 0.02 ^a^	0.38 ± 0.05 ^b^
Xanthones					
X1	Mangiferin	11.26 ± 1.98 ^b^	4.47 ± 0.41 ^a^	12.97 ± 2.63 ^b^	4.81 ± 0.62 ^a^
X2	Brasilixanthone B	n.d.	1.85 ± 0.23 ^a^	1.72 ± 0.10 ^a^	n.d.
X3	Trihydroxyxanthone-sulfonate	7.88 ± 0.97 ^b^	5.97 ± 0.34 ^a^	8.37 ± 0.66 ^b^	6.04 ± 0.40 ^a^
X4	Dimethylmangiferin	1.08 ± 0.12 ^b^	1.52 ± 0.21 ^b^	0.30 ± 0.02 ^a^	0.99 ± 0.19 ^b^
X5	Dihydroxy-metoxyxanthone-sulfonate	4.73 ± 0.38 ^b^	4.49 ± 0.44 ^ab^	3.50 ± 0.32 ^a^	3.94 ± 0.58 ^ab^
X6	Mangiferin *C*-prenyl isomer	0.18 ± 0.01 ^a^	n.d.	0.16 ± 0.05 ^a^	n.d.
X7	1,3,6,7-Tetrahydroxyxanthone 2-prenyl xanthone	n.d.	0.05 ± 0.00 ^a^	0.14 ± 0.02 ^b^	n.d.
X8	1,3,6,7-Tetrahydroxyxanthone 8-prenyl xanthone	n.d.	n.d.	0.14 ± 0.01	n.d.
X9	γ-Mangostin	0.62 ± 0.04 ^b^	0.47 ± 0.06 ^a^	0.48 ± 0.04 ^a^	0.78 ± 0.10 ^b^
X10	5-*O*-Methyl-2-deprenylrheediaxanthone B	n.d.	n.d.	0.21 ± 0.03 ^a^	0.15 ± 0.02 ^a^
X11	Cadensin G	6.83 ± 0.81 ^ab^	5.59 ± 0.77 ^a^	7.41 ± 0.46 ^b^	6.77 ± 0.61 ^ab^

* Phenolic compound contents are expressed as milligrams per gram dry weight (mg·g^−1^ DW±SD). NTS: non-transformed shoots; TSL B, TSL F, TSL H: transformed shoot lines; n.d.: not detected; S.D.: standard deviation. The values in one row marked with different lower-cases denote significant differences between samples at *p* < 0.05. The results for naphthodianthrone contents in tested shoot samples were previously published [25].

**Table 2 molecules-29-03893-t002:** Enzyme-inhibitory activities of *Hypericum perforatum* shoot cultures expressed as IC_50/25_ values *.

	MAO-A (IC_50_ µg·mL^−1^)	AChE (IC_50_ µg·mL^−1^)	BChE (IC_25_ µg·mL^−1^)	TYR (IC_50_ µg·mL^−1^)	α-AM (IC_25_ µg·mL^−1^)	α-GL (IC_50_ µg·mL^−1^)	PL (IC_50_ µg·mL^−1^)	CHE (IC_50_ µg·mL^−1^)
NTS	511.99 ± 51.70 ^bc^	1107.23 ± 78.76 ^d^	75.12 ± 10.56 ^b^	150.44 ± 10.12 ^d^	277.04 ± 30.15 ^d^	156.99 ± 9.78 ^b^	406.95 ± 49.64 ^c^	302.17 ± 1.91 ^c^
TSL B	433.90 ± 30.03 ^b^	944.91 ± 59.11 ^c^	480.03 ± 14.34 ^c^	80.66 ± 2.25 ^b^	261.98 ± 12.96 ^d^	345.69 ± 37.48 ^c^	352.68 ± 26.34 ^c^	125.93 ± 12.70 ^b^
TSL F	546.77 ± 47.03 ^c^	233.32 ± 25.82 ^b^	1257.49 ± 42.04 ^e^	147.59 ± 14.29 ^d^	214.43 ± 27.40 ^c^	879.90 ± 23.66 ^d^	230.39 ± 21.58 ^b^	572.04 ± 40.96 ^d^
TSL H	566.16 ± 41.01 ^c^	217.90 ± 17.12 ^b^	850.49 ± 51.92 ^d^	119.16 ± 9.77 ^c^	169.91 ± 7.52 ^b^	298.86 ± 30.79 ^c^	254.80 ± 24.79 ^b^	102.50 ± 6.50 ^a^
DCP	36.86 ± 0.70 ^a^	n.t.	n.t.	n.t.	n.t.	n.t.	n.t.	n.t.
Eserine	n.t.	11.97 ± 0.64 ^a^	0.10 ± 0.02 ^a^	n.t.	n.t.	n.t.	n.t.	n.t.
Kojic acid	n.t.	n.t.	n.t.	19.39 ± 1.77 ^a^	n.t.	n.t.	n.t.	n.t.
Acarbose	n.t.	n.t.	n.t.	n.t.	6.77 ± 0.76 ^a^	15.43 ± 1.15 ^a^	n.t.	n.t.
Orlistat	n.t.	n.t.	n.t.	n.t.	n.t.	n.t.	13.88 ± 0.77 ^a^	n.t.
Simvastatin	n.t.	n.t.	n.t.	n.t.	n.t.	n.t.	n.t.	160.47 ± 28.29 ^b^

* The values in one column marked with different lower-case letters denote significant differences at *p* < 0.05 between clones. IC_50_: extract concentration that inhibits 50% of enzyme activity; IC_25_: extract concentration that inhibits 25% of enzyme activity. NTS: non-transformed shoots; TSL B, TSL F, TSL H: transformed shoot lines; MAO-A: monoamine oxidase-A; AChE: acetylcholinesterase; BChE: butyrylcholinetserase; TYR: tyrosinase; α-AM: α-amylase; α-GL: α-glucosidase; PL: pancreatic lipase; CHE: cholesterol esterase; DCP: 2,4-dichlorophenol; n.t.: not tested.

**Table 3 molecules-29-03893-t003:** Binding energy and inhibition constant of the best-ranked docking pose of selected ligands and enzymes *.

Ligands	Enzymes	Binding Energy (kcal·mol^−1^)	Inhibition Constant (K_i_)
Chlorogenic acid	MAO-A	−7.31	4.4 µM
	AChE	−7.30	4.49 µM
	BChE	−5.84	52.07 µM
	TYR	−8.09	1.17 µM
	α-AM	−4.30	703.86 µM
	α-GL	−5.09	156.36 µM
	PL	−6.14	31.74 µM
	CHE	−5.02	208.55 µM
Epicatechin	MAO-A	−8.52	564.15 nM
	AChE	−8.19	995.35 nM
	BChE	−8.45	635.22 nM
	TYR	−4.37	622.42 µM
	α-AM	−7.76	2.04 µM
	α-GL	−7.44	3.51 µM
	PL	−8.38	722.92 nM
	CHE	−5.64	73.02 µM
Pseudohypericin	MAO-A	−6.50	17.15 µM
	AChE	−12.00	1.59 nM
	BChE	−14.56	21.14 pM
	TYR	−7.21	5.20 µM
	α-AM	−11.58	3.27 nM
	α-GL	−11.65	2.89 nM
	PL	−12.70	489.00 pM
	CHE	−10.51	19.91 nM
Hyperforin	MAO-A	−4.99	270.76 µM
	AChE	−8.49	600.06 nM
	BChE	−11.06	7.75 nM
	TYR	−6.37	21.33 µM
	α-AM	−8.02	1.33 µM
	α-GL	−10.30	28.32 nM
	PL	−8.42	673.63 nM
	CHE	−4.48	518.91 µM

* MAO-A: monoamine oxidase-A; AChE: acetylcholinesterase; BChE: butytylcholinesterase; TYR: tyrosinase; α-AM: α-amylase; α-GL: α-glucosidase; PL: pancreatic lipase; CHE: cholesterol esterase.

## Data Availability

Data are contained within the article.

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
