# Peer review of "Phytochemical Analysis, Biological Activities, and Docking of Phenolics from Shoot Cultures of Hypericum perforatum L. Transformed by Agrobacterium rhizogenes"

_molecules, 2024, doi:10.3390/molecules29163893_

Round 1

Reviewer 1 Report

Comments and Suggestions for Authors

The article submitted by Tusevski et al. concerns the study of substances contained in transformed roots of H. perforatum.

St. John's wort as a plant containing bioactive compounds, despite many years of analysis, still arouses the interest of scientists, so the object of the research is most legitimate. The methods chosen also seem sensible, but the article itself leaves a bit of understatement raising questions. 

The roots were divided into 3 groups B, F and H, but no explanation of what they mean, how did they differ? This should be a starting point for a deep discussion of the compositional differences between them.

Was the determination of the substance content of the roots validated? In how many replicates was the determination performed? 

In how many repetitions were biolgoic activity tests performed?

On what basis were the protein structures selected for docking? Why were some of them derived from organisms other than humans?

What was the optimisation of the phytoligands based on?

How many ligand conformations were generated as part of the docking simulation?

Was an attempt made to validate the molecular docking simulation?

The article needs editorial changes:

The tables with compound or docking results could be shortened, the most important results selected and the rest moved to supplementary materials.

Abbreviations should be explained when they first appear - currently this has to be sought further down in the text (e.g. NTS)

Discussion of the constant Ki inhibition from docking is missing.

There is some surprise in the selection of literature that as many as 14 items, which is more than 20% of the items, are publications by the first author. I understand the need to refer to previous studies, but in this case the number seems somewhat excessive.

Reviewer 2 Report

Comments and Suggestions for Authors

Dear Authers,

The manuscript described the chemical analysis, biological activities and docking of the phenolics from shoot cultures of Hypericum perforatum transformed by Agrobacterium rhizogenes. The results showed that Agrobacterium rhizogenes-mediated transformed shoot cultures of Hypericum perforatum should have their potential as an efficient biotechnological system for production of bioactive phenolics with pharmaceutical applications. However, the revision is necessary before acceptance. Some comments and suggestions are listed below.

1. I prefer the article title as: Analysis, biological activities and docking of the phenolics from shoot cultures of Hypericum perforatum transformed by Agrobacterium rhizogenes.

2. p.2, line 69: “Rhizobium rhizogenes” should be “Agrobacterium rhizogenes”.

3. p.2, line 74: “R. rhizogenes” should be “A. rhizogenes”. Please check other plances.

4. If it is possible, please provide the structures of the 31 compounds listed in Table 1.

Reviewer 3 Report

Comments and Suggestions for Authors

This article is relatively reasonable and reliable. In this article, Hypericum transformed shootlines regenerated from corresponding hairy roots were evaluated for phenolic compounds contents and in vitro inhibitory potential against enzymes associated with depression, neurodegeneration, diabetes and obesity. The review’s logic is rigorous, and the data of the experiments are convincing. This article inhibited that the transform shoots of H. perforatum can be an excellent biotechnological system for producing phenolic compounds. It is a topic of interest to the researchers in the related areas, especially for the the significant antioxidant activities article, but still has some problems.

1. The H-bond of pseudohypericin with α-amylase are in gate keeper whther or not? Please describe it.

2. Why chose the phenolic compounds of H. perforatum as study target? Please add the detailed description.

3. How about the cultured effiency and the transformation ratio of TSL B, TSL F, TSL H ?

4. The spectrum of compounds should be add.

5. The abstract was too cumbersome. Its unworthy too spend many words to describe the details of experiment. Abstract should focus on logic.

Round 2

Reviewer 1 Report

Comments and Suggestions for Authors

The Authors have improved the quality of manuscript. The answered all questions and suggestions. In my opinion the article now can be accepted for publication.

Reviewer 2 Report

Comments and Suggestions for Authors

The authors have addressed all my questions, and I have no additional suggestion.